# Determinants of photoplethysmography signal quality at the wrist

Peter H. Charlton[1,2]*, Vaidotas Marozas[3,4], Elisa Mejía-Mejía[5], Panicos A. Kyriacou[2], Jonathan Mant[1]

**1** Department of Public Health and Primary Care, University of Cambridge, Cambridge, United Kingdom, **2** Research Centre for Biomedical Engineering, City, University of London, London, United Kingdom, **3** Biomedical Engineering Institute, Kaunas University of Technology, Kaunas, Lithuania, **4** Faculty of Electrical and Electronics Engineering, Kaunas University of Technology, Kaunas, Lithuania, **5** Rush Alzheimer's Disease Center, Rush University Medical Center, Chicago, Illinois, United States of America

* pc657@cam.ac.uk

**Data availability statement:** The Aurora-BP dataset used in this study is freely available to approved users as detailed here: https://github.com/microsoft/aurorabp-sample-data.

## Abstract

Wrist photoplethysmogram (PPG) signals are widely used for physiological monitoring in consumer devices. However, the PPG is highly susceptible to noise, which can reduce the accuracy of monitored parameters. The aim of this study was to identify factors which influence PPG signal quality. Data from the Aurora-BP dataset were used, consisting of reflectance wrist PPG signals measured from 1,142 subjects of varying ages and health statuses. Measurements were acquired in supine, sitting, and standing postures, and with the sensor held at different heights. Three signal quality metrics were calculated: the signal-to-noise ratio (SNR), the perfusion index (PI), and the template-matching correlation coefficient (TMCC). When comparing between postures with the sensor held at a natural height, quality was greatest in the supine position (SNR: 18.6 dB), followed by sitting with the arm resting in the lap (13.7 dB), and lowest whilst standing with the arm hanging alongside (9.0 dB) (p<0.001). Signal quality increased as the arm was raised to heart height: whilst sitting, quality was lowest with the arm alongside the body (10.5 dB), and increased when the sensor was held in the lap (13.7 dB) and at heart height (15.5 dB) (p<0.001). Similar trends were observed for the TMCC and PI. Findings were mixed for the influence of participant characteristics on signal quality. The SNR and TMCC, but not the PI, increased with age. The SNR either decreased or remained constant at darker skin tones when controlling for PPG DC amplitude, compared to constant or increased when allowing DC amplitude to vary. In conclusion, this study identified the impacts of posture and sensor height on signal quality, with highest qualities observed in the supine posture and with the sensor at heart height. It also highlights the importance of adjusting LED light intensity to maintain signal quality across skin tones.

The code used for data analysis is available at: https://github.com/peterhcharlton/PPG_quality_determinants.

**Funding:** This work was supported by: the British Heart Foundation grant (FS/20/20/34626 to PHC) (https://www.bhf.org.uk/); and the European Partnership on Metrology (22HLT01 to PHC and VM) (https://www.euramet.org/). The project (22HLT01 QUMPHY) has received funding from the European Partnership on Metrology, co-financed from the European Union's Horizon Europe Research and Innovation Programme and by the Participating States. Funding for the University of Cambridge was provided by Innovate UK under the Horizon Europe Guarantee Extension, grant number 10091955. The funders had no role in study design, data collection and analysis, decision to publish, or preparation of the manuscript.

**Competing interests:** PHC has performed consultancy work for Cambridge University Technical Services, has received travel funds from VascAgeNet, and has received honoraria from IOP Publishing and Emory University (the latter not received personally). JM has performed consultancy work for BMS/Pfizer, Omron, and Novonordisk.

## Author summary

Photoplethysmography sensors are widely used in devices such as smartwatches and fitness trackers. The photoplethysmogram (PPG) signal can be used to obtain several physiological measurements such as heart rate and blood oxygen saturation. However, the PPG signal is highly susceptible to noise which can lead to inaccurate measurements. The aim of this study was to identify factors which influence PPG signal quality. We analysed measurements from over 1,000 subjects collected in laboratory conditions using wrist-worn sensors similar to smartwatches. We found that signal quality was highest when subjects were lying down, and signal quality degraded when sitting and standing. In addition, we found that signal quality was highest when the sensor was at heart height, and degraded when at lower heights such as with the arm hanging alongside the body. We also found that automatic adjustment of the light intensity of the sensor may help maintain high signal quality across different skin tones. These findings provide insight into how to obtain high quality PPG signals in daily life, which may help improve the accuracy of wearable data.

## Introduction

Photoplethysmography is a physiological sensing modality which enables non-invasive measurement of the arterial pulse. The resulting photoplethysmogram (PPG) signal exhibits a pulse wave for each heart beat, and contains a wealth of information on the heart and vasculature [1]. Photoplethysmography is used in clinical devices such as pulse oximeters [2], and consumer devices such as smartwatches [3]. It is widely used for heart rate and oxygen saturation monitoring, and a range of additional applications are in development including blood pressure monitoring, assessment of vascular health, identification of peripheral arterial disease, and detection of obstructive sleep apnea [4]. However, the PPG signal is highly susceptible to noise [5] which can render signals unsuitable for analysis [4] or lead to inaccurate measurements [6]. Therefore, it is important to understand the factors which cause noise in the PPG signal in order to mitigate against them.

The level of noise in a PPG signal is often referred to as 'signal quality', where high-quality signals contain relatively little noise and low-quality signals are highly corrupted by noise. Examples of low-, medium- and high-quality signals are shown in Fig 1. PPG signal quality has always been an important consideration, whether in transmission photoplethysmography used in pulse oximeters [7], reflectance photoplethysmography used in wearables [8] or smartphones [9], or remote photoplethysmography acquired using cameras [10]. The issue of PPG signal quality is becoming increasingly important with the widespread use of photoplethysmography in wearables, which acquire PPG signals in challenging activities without manual inspection of the recording technique or resulting signal quality. The issue of low PPG signal quality can be addressed by: (i) identifying and excluding periods of low-quality signals from analysis [11], which can be achieved using classical signal processing algorithms and machine learning [12] or neural network algorithms [13]; and enhancing signal quality through digital filtering or motion artifact removal [14], which can benefit from incorporating information about movement from simultaneous accelerometry signals [15]. While various methods can mitigate low signal quality, it is important to understand how to optimise PPG signal acquisition, as low-quality signals cannot always be fully corrected through post-processing.

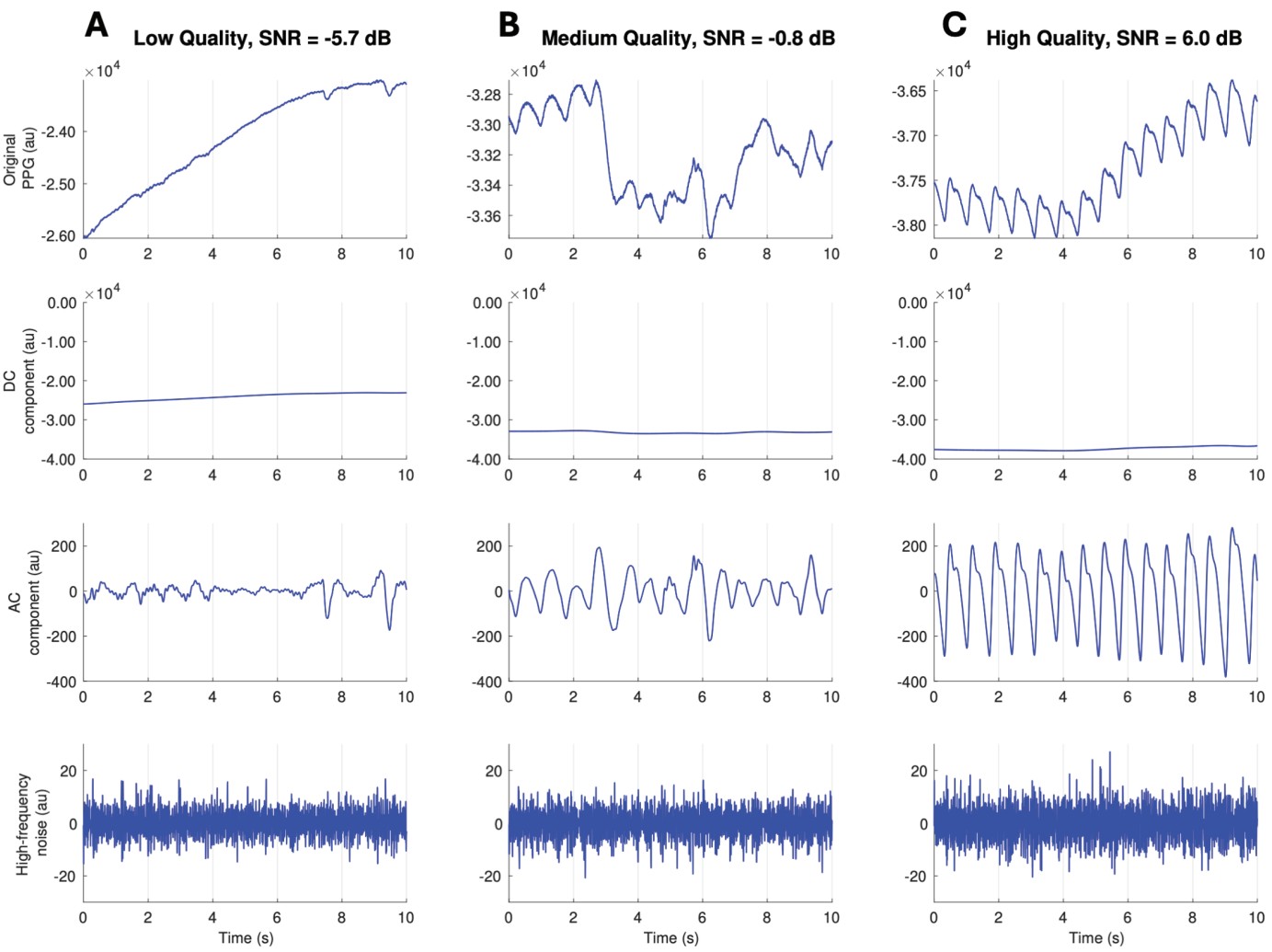

**Fig 1. Examples of PPG signals**: Each column shows an original PPG signal (scaled to fit) in the top row, and individual components of this PPG signal in the subsequent rows, each shown on common scales: (second row) the DC component (low-pass filtered below 0.5 Hz); (third row) the AC component (band-pass filtered between 0.5 and 12 Hz); and (fourth row) high-frequency noise (high-pass filtered above 12 Hz). Each column shows a different signal and its components: (a) a low-quality PPG signal with low-amplitude AC components and barely discernible pulse waves; (b) a medium-quality PPG signal where individual pulse waves are visible but their shapes are corrupted by noise; and (c) a high-quality PPG signal where individual pulse waves as well as their detailed shapes are clearly visible. *Definition: au - arbitrary units.*

Several factors are known to influence PPG signal quality, with key studies relating to each factor summarised in Table 1. First, the sensor design can influence PPG signal quality by affecting the intensity of light at a tissue region (e.g., via LED light intensity or LED viewing angle), and the optical path taken between the LED and detector (e.g., via sensor geometry, source-detector distance, or the wavelength of light). Second, the position and mode of sensor attachment can influence PPG signal quality by affecting the quality of sensor-skin contact and the level of pulsatility in blood volume (e.g., via the contact pressure applied, and the height of the sensor relative to the heart). Of particular relevance to this study, the use of higher measurement sites has been found to either increase the amplitude of the pulsatile component of the PPG signal [16–18] or result in only small changes in amplitude [19]. Third, subject characteristics can influence PPG signal quality by affecting the amount of blood flow

**Table 1. Factors influencing PPG signal quality.**

| Factor | Details |
|---|---|
| *Sensor design* | |
| LED light intensity | The amplitude of the PPG signal increases with LED light intensity [20], corresponding to a higher signal quality. |
| LED viewing angle | The viewing angle (indicating the angular range over which emitted light is visible) affects PPG signal quality, with smaller angles providing a narrower focus and producing a higher signal quality [21]. |
| Multiple signals | Acquisition of multiple PPG signals by a single sensor, such as by having multiple LEDs and/or detectors to provide, can improve the quality of a composite signal derived through (weighted) averaging [22,23] or signal decomposition [24]. |
| Sampling frequency | Inadequate sampling frequency can distort PPG pulse wave morphology [25]. |
| Sensor geometry | The positioning of the LED(s) and detector(s) relative to each other affects PPG signal magnitude, particularly when using shorter wavelengths such as red light [23]. Ideally, the light source should surround the detector, and an optical barrier should be included to block scattered light [23,26]. |
| Signal filtering | High-pass cut-off frequencies of 0.5 Hz and above can distort PPG pulse wave morphologies [27]. |
| Source-detector distance | The distance between the light source (LED) and the light detector influences PPG signal quality, with greater distances resulting in lower signal-to-noise ratios [28] and higher AC/DC ratios [26]. |
| Wavelength of light | Studies comparing different light wavelengths have found that: (i) when compared with red, blue or green LEDs, yellow LEDs provided higher measured light intensities [29], and yellowish-green LEDs provided higher signal-to-noise ratios [28]; (ii) green light provided a greater AC/DC ratio than blue, red, or infrared light [30], and was less affected by extreme (high or low) temperatures than infrared light [31]. |
| *Sensor attachment* | |
| Contact pressure | Wrist PPG AC amplitude increased with increasing contact pressure [32]. Finger PPG AC amplitude was found to be highest within an optimal range of contact pressures [33]. Optimal contact forces may differ between applications [34]. |
| Sensor height relative to heart | Signal quality decreases at lower sensor heights relative to the heart [18]. |
| Sensor-skin contact | Gaps between the sensor (either LED or photodiode) and the skin resulted in poorer signal quality [35]. Flexible, conformal sensors can help improve signal quality during motion [36,37]. |
| *Subject characteristics* | |
| Body mass index (BMI) | The BMI indirectly influences the PPG signal by altering the dermal capillary density and its depth, as well as modifying the skin thickness and the trans-epidermal water loss. These changes lead to variations in the path and properties of the tissues that the light passes through [38]. |
| Perfusion | PPG signal quality is closely associated with perfusion (levels of blood flow), since the PPG is strongly influenced by pulsatile blood flow. The 'perfusion index' is calculated as the AC/DC ratio [39], which is also used to assess PPG signal quality. A range of factors have been found to be associated with the perfusion index [39]. |
| Skin tone | PPG signal quality is lower in darker skin tones [30]. This may affect the reliability of pulse oximeters [40]. |
| *Recording setting* | |
| Ambient lighting | Static and variable levels of ambient light can interfere with the PPG signal by contributing to the DC and AC components respectively [41]. |
| External temperature | Signal quality is lower at lower temperatures due to reduced perfusion [42–44]. |
| Movement | Movement causes motion artifact, a noise with a broad frequency spectrum overlapping that of PPG pulse waves [5]. Higher intensity activities result in worse signal quality [45]. |
| Nighttime | Signal quality is higher whilst sleeping at night [45–47]. |
| Posture | Small changes in DC intensity were observed when changing from sitting to standing positions [19]. |

(e.g., perfusion) and the passage of light through the skin (e.g., skin tone). Fourth, the recording setting can influence PPG signal quality by affecting the blood volume (e.g., external temperature), the level of ambient light, and the presence of motion artifact (e.g., movement vs. nighttime recordings). It is important to understand how these factors influence PPG signal quality to optimise PPG signal acquisition. Whilst much is known about some factors (e.g., the detrimental effect of movement on PPG signal quality), relatively little is known about others (e.g., the effects of skin tone, sensor height relative to the heart, and posture).

The aim of this study was to identify factors which influence reflectance PPG signal quality, and assess their relative contributions to the signal quality. The study was performed using a large dataset of wrist PPG signals acquired in laboratory settings. The results provide insight into how to optimise PPG signal acquisition for physiological monitoring in daily life.

## Methods

### Ethics Statement

Ethical approval was not required for this study as it used pre-existing, anonymised data.

### Dataset

The Aurora-BP dataset was used in this study [48]. This dataset was originally used to compare different wearable signals for cuffless blood pressure estimation, namely tonometry, electrocardiography, and photoplethysmography signals [48]. The dataset consists of two data subsets collected from different participants using different protocols: the Auscultatory and Oscillometric Subsets. These subsets have different characteristics and were used for different purposes in the current study. The Auscultatory Subset contains skin colour data for all participants, whereas the Oscillometric Subset only contains skin colour data for 29% of participants. Therefore, the Auscultatory Subset was primarily used to investigate the impact of participant characteristics on PPG signal quality, namely: age, gender, the presence of diabetes, skin color, and blood pressure. The Oscillometric Subset contains measurements taken in three postures (supine, sitting, and standing) and at three different sensor heights ("arm down"—hanging alongside body; "arm in lap"; and "arm up" at heart height), whereas the Auscultatory Subset only contains measurements taken in two postures (supine and sitting) at a single sensor height ("arm up"). Therefore, the Oscillometric Subset was primarily used to investigate the impact of posture and sensor height on PPG signal quality. The Aurora-BP dataset is freely available to approved users, as described in [48] and documented at https://github.com/microsoft/aurorabp-sample-data (from where much of the following information is obtained).

The data collection and selection processes are now described. In both protocols participants had measurements taken in two laboratory visits (an initial and a return visit), and in addition in the oscillometric protocol measurements were also obtained between visits. We used data from the initial visits in this study. During the auscultatory protocol data were collected whilst supine and seated in the initial visit, although seated data were only collected for a subset of participants. Therefore, we only used supine data from the Auscultatory Subset. A total of six supine measurements were obtained, each lasting approximately 20 s, as detailed in Table 2. During the oscillometric protocol data were collected whilst sitting, supine and standing, as detailed in Table 2. Separate measurements were obtained at different sensor heights whilst sitting (arm hanging alongside body, hand in lap, and at heart height) and whilst standing (arm hanging alongside body, and at heart height). Each measurement lasted approximately 30 s. For both protocols, each measurement was taken at least 60 s apart, with a "five-minute rest in-position for each postural change." For this analysis, we only included participants who had complete datasets, with all measurements present and no missing signals (where any of the PPG, electrocardiogram, or accelerometry signals were missing as shown by a 'flat-line' appearance). This resulted in data being included for 643 out of the original 672 participants in the Auscultatory Subset, and 499 out of the original 548 participants in the Oscillometric Subset. The characteristics of the included participants are summarized in Table 3.

PPG signals were measured on the anterior side of the forearm using the MAX30101 sensor (Maxim Integrated, San Jose, USA). PPG signals were acquired using the sensor's green LED, which has a wavelength of 537 nm [49]. In the auscultatory protocol, blood pressure was measured by two observers independently. Only systolic and diastolic blood pressures with

**Table 2. Measurement protocols.**

| Protocol stage | Posture | Sensor height | Duration (s), median (quartiles) | No. repetitions |
|---|---|---|---|---|
| *Auscultatory Protocol* | | | | |
| Calibration_start | supine | alongside body | 20.5 (18.4–23.3) | 3 |
| Static_challenge_start | supine | alongside body | 19.7 (18.0–22.3) | 3 |
| *Oscillometric Protocol* | | | | |
| Supine | supine | alongside body | 30.0 (30.0–30.0) | 2 |
| Sitting_arm_down | sitting | hanging alongside body | 30.0 (30.0–30.0) | 1 |
| Sitting_arm_lap | sitting | hand in lap | 30.0 (30.0–30.0) | 1 |
| Sitting_arm_up | sitting | at heart height | 30.0 (30.0–30.0) | 1 |
| Standing_arm_down | standing | hanging alongside body | 30.0 (30.0–30.0) | 1 |
| Standing_arm_up | standing | at heart height | 30.0 (30.0–30.0) | 1 |

**Table 3. Characteristics of participants included in the analysis.**

| Characteristic | Value | |
|---|---|---|
| | **Auscultatory Subset** | **Oscillometric Subset** |
| No. participants | 643 | 499 |
| Female, n (%) | 317 (49.3) | 243 (48.7) |
| Age (years), mean (SD) | 45.0 (13.0) | 45.5 (8.9) |
| BMI (kgm$^{-2}$), mean (SD) | 28.7 (6.6) | 30.5 (7.1) |
| Systolic blood pressure (mmHg), mean (SD) | 129 (19) | 131 (17) |
| Diastolic blood pressure (mmHg), mean (SD) | 78 (13) | 86 (11) |
| Skin color (Fitzpatrick scale), n (%): | | |
| 1 | 291 (45.3) | 8 (1.6) |
| 2 | 219 (34.1) | 60 (12.0) |
| 3 | 82 (12.8) | 48 (9.6) |
| 4 | 27 (4.2) | 16 (3.2) |
| 5 | 21 (3.3) | 6 (1.2) |
| 6 | 3 (0.5) | 3 (0.6) |
| Unknown | 0 (0.0) | 358 (71.7) |
| Self-reported history of, n (%): | | |
| managed hypertension | 127 (19.8) | 147 (29.5) |
| unmanaged hypertension | 95 (14.8) | 90 (18.0) |
| high blood pressure | 200 (31.1) | 214 (42.9) |
| diabetes | 54 (8.4) | 56 (11.2) |
| arrythmia | 21 (3.3) | 12 (2.4) |
| previous stroke | 10 (1.6) | 8 (1.6) |
| previous heart attack | 6 (0.9) | 6 (1.2) |
| coronary artery disease | 9 (1.4) | 4 (0.8) |
| heart failure | 2 (0.3) | 1 (0.2) |
| aortic stenosis | 0 (0.0) | 1 (0.2) |
| valvular heart disease | 2 (0.3) | 1 (0.2) |
| other cardiovascular diseases | 47 (7.3) | 58 (11.6) |
| currently taking medicine for cardiovascular disease | 158 (24.6) | 195 (39.1) |
| No self-reported cardiovascular disease or medication | 373 (58.0) | 218 (43.7) |
| DC amplitude (arbitrary units), n (%): | | |
| 0k-50k | 148 (23.0) | 341 (68.3) |
| 295k-330k | 264 (41.1) | 138 (27.7) |
| 570k-605k | 202 (31.4) | 19 (3.8) |
| 850k-900k | 29 (4.5) | 0 (0.0) |
| 1135k-1140k | 0 (0.0) | 1 (0.2) |

an inter-observer difference of <10 mmHg were included in this analysis. Pulse pressure was subsequently calculated from acceptable systolic and diastolic pressures.

## PPG signal processing

PPG signals were processed using three different approaches to obtain three parameters indicative of signal quality: (i) the signal-to-noise ratio (SNR); (ii) the perfusion index (PI, the ratio of the amplitudes of the pulsatile (AC) and baseline (DC) components, also known as the AC/DC ratio); and (iii) a template-matching correlation coefficient (TMCC). These three approaches are now described.

The first processing approach consisted of filtering the signals to eliminate low-frequency variations below the minimum plausible cardiac frequency, and then calculating the SNR. The SNR represented the ratio of the spectral power in the cardiac frequency and its harmonics (*i.e.* associated with heart rate) to that of the remaining frequencies which represent noise. The filtering was designed to ensure that the frequency with the highest spectral power (the fundamental frequency) was the cardiac frequency as opposed to any low frequency variations. A zero-phase Chebyshev II bandpass filter was used with order 4 (as recommended in [50]), high-pass frequency cut-off of 0.5 Hz (corresponding to 30 beats per minute, bpm), and low-pass frequency cut-off of 12 Hz. The filter was designed using Matlab's `cheby2` function (Matlab 2022a, The Mathworks Inc.). The signal-to-noise ratio was then calculated as the ratio of the power in the fundamental frequency (defined as the spectral peak surrounding the fundamental frequency with all values monotonically decreasing around this frequency) to the power in the noise (defined as all the remaining frequency components apart from the DC component and harmonics of the fundamental frequency). Matlab's `snr` function was used to calculate the SNR as follows:

$$\text{SNR (dB)} = 10 \log_{10} \left( \frac{P_{\text{fundamental}}}{P_{\text{noise}}} \right), \tag{1}$$

where $P_{\text{fundamental}}$ and $P_{\text{noise}}$ are the power in the fundamental frequency and the power in the noise respectively.

The second approach consisted of filtering the signals to ensure accurate beat detection, detecting individual pulse waves, and calculating the AC and DC amplitudes and PI for each pulse in the original signal. To do so, PPG signals were filtered with a zero-phase, band-pass Butterworth filter with order 4, high-pass frequency cut-off of 0.5 Hz, and low-pass frequency cut-off of 8.0 Hz. These cut-off frequencies are similar to the 0.67 and 8.0 Hz cut-offs used for beat detection in [51], with a slight widening of the band-pass region to ensure the lowest heart rates are retained for analysis. This filtered signal was used for beat detection. Individual pulse waves were detected using the multi-scale peak trough detector (MSPTD) beat detector [52], which has previously been found to be one of the best-performing open-source beat detection algorithms for wrist PPG signals [51], and is openly available in the `ppg-beats` Matlab library [53]. Having identified individual pulse waves, their onsets and peaks were located (as the minimum and maximum values between successive beat detections), and the mid-points on the systolic upslope of PPG pulse waves were detected. The pulsatile (AC) component was calculated as the difference between the amplitudes of onsets and peaks in the original signal (as opposed to the filtered signal), and the baseline (DC) component was calculated as the absolute amplitudes of mid-points of systolic upslopes, once again in the original signal. The PI was calculated from the median AC and DC amplitudes,

$$\text{PI (\%)} \quad = \quad 100 \times \frac{\text{AC amplitude}}{\text{DC amplitude}}. \tag{2}$$

The AC and DC components of the PPG signal are illustrated in Fig 2.

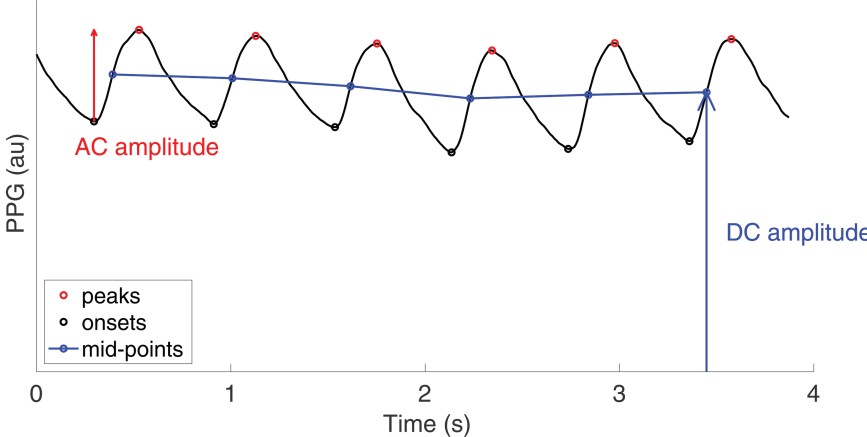

**Fig 2. The AC and DC components of the PPG signal.**

The third approach consisted of finding the correlation between each individual pulse wave and a template ('average') pulse wave [54]. To do so, the median inter-beat interval was calculated, and the PPG signal was segmented into individual pulse waves centred on the mid-points with a window duration of the median inter-beat interval. A template pulse wave was calculated by aligning all the individual pulse waves, and finding the mean value at each sample point. Correlation coefficients were calculated between this template and each individual pulse wave, and the final TMCC was calculated as the mean of the correlation coefficients for each individual pulse wave. The approach of calculating the template pulse wave from all individual pulse waves, regardless of their quality, was chosen to ensure a quantitative assessment of signal quality could be derived from all input signals regardless of their quality. This approach has been used previously [54,55], although we note that others have proposed only using high-quality pulse waves when calculating the template [56,57].

Higher values indicate higher signal quality for all three metrics (SNR, PI, and TMCC).

## Statistical analysis

The observed levels of signal quality were reported using non-parametric statistics (median and inter-quartile range), since not all variables were normally distributed (namely the PI and TMCC). Parametric statistics (mean and standard deviation, SD) were also reported for comparison with the literature.

Linear mixed effects modelling was used to investigate the impact of various factors on PPG signal quality [58]. The `fitlme` function in Matlab (R2022a, Mathworks, Natick, USA) was used, with maximum likelihood estimation used to estimate model parameters. First, the impact of posture and sensor height on PPG signal quality was assessed using direct comparisons on the Oscillometric Subset. Since each participant in this subset contributed the same number of recordings at each posture and sensor height, it was possible to perform direct comparisons of PPG signal quality between different postures and sensor heights. To do so, a mixed effects model was fitted for each signal quality metric, with posture (or sensor height) specified as a fixed effect, and subject-specific intercepts included as random effects. These models were used to test for significant differences between different postures and sensor heights at the 5% significance level, using the p-value from the model's fixed-effects coefficient estimates. A total of 30 tests were performed (10 tests for each of the 3 signal quality metrics).

Therefore, a Holm-Sidak correction was used to correct for multiple comparisons [59,60], as implemented in [61]. Data were presented in boxplots showing the median, lower and upper quartiles, and whiskers representing the range of the non-outlying data (where outliers were defined as those data lying more than 1.5 times the inter-quartile range from the lower and upper quartiles).

Second, mixed effects modelling was used to investigate the impact of a range of factors on PPG signal quality [58]. This was performed on the Auscultatory and Oscillometric Subsets individually. The following participant characteristics were entered into the model as fixed effects (without interaction terms): age, gender, body mass index (BMI), the presence of diabetes, skin colour, systolic blood pressure, pulse pressure, and DC amplitude. In addition, when analysing the oscillometric dataset, posture and sensor height were also included as fixed effects (where supine, sitting, and standing postures were included, and 'arm down', 'arm in lap', and 'arm up' sensor heights). In a second model the additional fixed effect of DC amplitude was excluded, representing a device characteristic. Systolic blood pressure was chosen rather than diastolic blood pressure because it was found to have a slightly stronger association with PPG signal quality. In both models, we included subject-specific intercepts as random effects to model variability in baseline signal quality between subjects which was not explained by the participant characteristics (e.g., variability due to subject-specific differences in blood flow at the wrist). It was possible to account for subject-specific differences at baseline because of the repeated measures per subject. However, we did not include subject-specific random slopes because the number of repeated measures per subject was considered too small to account for this (e.g., to account for differences between subjects in how a fixed effect such as posture influenced signal quality). The use of two models (with and without DC amplitude) allowed us to investigate the potential benefit of varying DC amplitude to reduce the impact of participant characteristics on signal quality.

## Results

### Observed levels of PPG signal quality

The observed levels of PPG signal quality are summarized in Table 4 and Fig 3. SNRs ranged from –7.5 to 60.2 dB, with typical values of approximately 10 to 20 dB. PIs ranged from 0.00 to 6.61%, with typical values of approximately 0.1 to 0.8%. TMCCs ranged from –0.18 to 1.00, with typical values of approximately 0.8 to 1.0. The three signal quality metrics provided quite different information, as shown by low correlations between them: $R^2$ values on the Auscultatory and Oscillometric Subsets respectively of 0.02 and 0.12 (PI vs. TMCC), 0.12 and 0.16 (SNR vs. PI), and 0.13 and 0.36 (SNR vs. TMCC). It should be noted that the signal quality metrics are not perfect indicators of signal quality. For instance, Fig 4 shows examples of PPG signals whose SNR values did not appear to align with their quality.

### Influence of posture on PPG signal quality

The influence of posture on PPG signal quality is demonstrated in Fig 5. The upper row demonstrates that signal quality differed greatly between postures when the arm was held in a natural position: 9.0 (3.7–13.2) dB when the arm was alongside the body whilst standing; 13.7 (8.2–17.8) dB when the arm was in the lap whilst sitting; and 18.6 (14.5–22.6) dB when the arm was at approximately heart height whilst supine. When the sensor was held at heart height in all three postures, PPG signal quality was highest in the supine position (SNR: 18.6 (14.5–22.6) dB), followed by the standing position (16.6 (12.2–20.3) dB), followed by the sitting position (15.5 (11.3–19.2) dB) (all p<0.001). This was true when assessing quality using

**Table 4. The observed levels of PPG signal quality.**

| Protocol stage | SNR (dB) | | PI (%) | | TMCC | |
|---|---|---|---|---|---|---|
| | median (quartiles) | mean (SD) | median (quartiles) | mean (SD) | median (quartiles) | mean (SD) |
| *Auscultatory Subset* | | | | | | |
| calibration_start | 17.1 (11.3–23.3) | 17.4 (8.7) | 0.12 (0.07–0.26) | 0.26 (0.35) | 0.97 (0.91–0.99) | 0.93 (0.10) |
| static_challenge_start | 18.7 (12.5–25.2) | 18.8 (8.8) | 0.14 (0.08–0.30) | 0.30 (0.41) | 0.97 (0.92–0.99) | 0.93 (0.10) |
| *Oscillometric Subset* | | | | | | |
| sitting_arm_down | 10.5 (5.4–15.2) | 10.4 (7.3) | 0.20 (0.08–0.42) | 0.31 (0.33) | 0.85 (0.70–0.94) | 0.79 (0.18) |
| sitting_arm_lap | 13.7 (8.2–17.8) | 12.9 (6.7) | 0.31 (0.11–0.64) | 0.45 (0.48) | 0.93 (0.81–0.97) | 0.87 (0.15) |
| sitting_arm_up | 15.5 (11.3–19.2) | 15.1 (6.1) | 0.38 (0.13–0.74) | 0.52 (0.53) | 0.95 (0.90–0.98) | 0.92 (0.10) |
| standing_arm_down | 9.0 (3.7–13.2) | 8.6 (6.7) | 0.18 (0.08–0.37) | 0.28 (0.31) | 0.82 (0.67–0.92) | 0.76 (0.20) |
| standing_arm_up | 16.6 (12.2–20.3) | 16.1 (6.3) | 0.49 (0.18–0.99) | 0.71 (0.71) | 0.96 (0.91–0.99) | 0.92 (0.11) |
| supine | 18.6 (14.5–22.6) | 18.3 (6.3) | 0.76 (0.25–1.44) | 0.99 (0.92) | 0.99 (0.96–0.99) | 0.96 (0.07) |

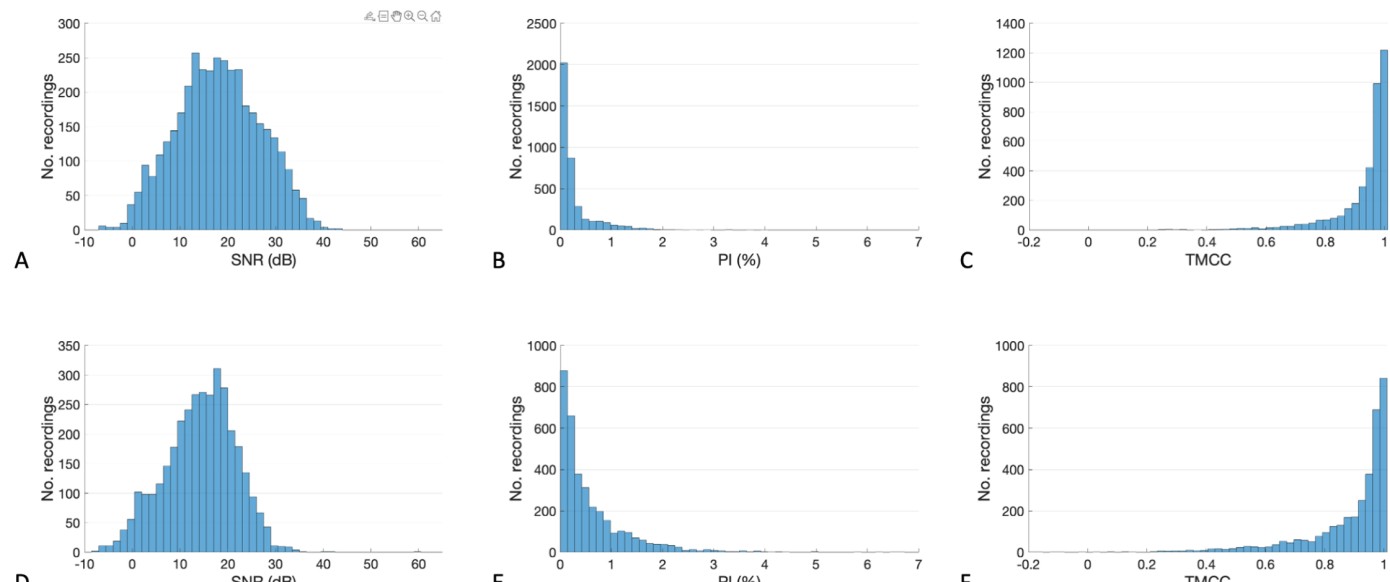

**Fig 3. PPG signal quality metrics**: The distributions of PPG signal quality metrics in: (upper row, A–C) the Auscultatory Subset; and (lower row, D–F) the Oscillometric Subset.

all the signal quality metrics, apart from one non-significant difference between TMCC's between sitting and standing (Fig 5F).

## Influence of sensor height on PPG signal quality

The influence of sensor height on PPG signal quality is demonstrated in Fig 6. The results show that PPG signal quality was significantly increased at higher sensor heights. This was true for both standing and sitting postures (upper and lower rows respectively), and when assessing signal quality using all the signal quality metrics (left, middle, and right hand panels respectively). When standing, the signal quality was increased at the "arm up" sensor height (i.e., sensor held up at heart height) compared to the 'arm down' sensor height (i.e., arm hanging down alongside the body) (SNR 16.6 (12.2–20.3) dB vs. 9.0 (3.7–13.2) dB, p<0.001). Similarly, signal quality was increased at higher sensor heights in the sitting posture (SNR

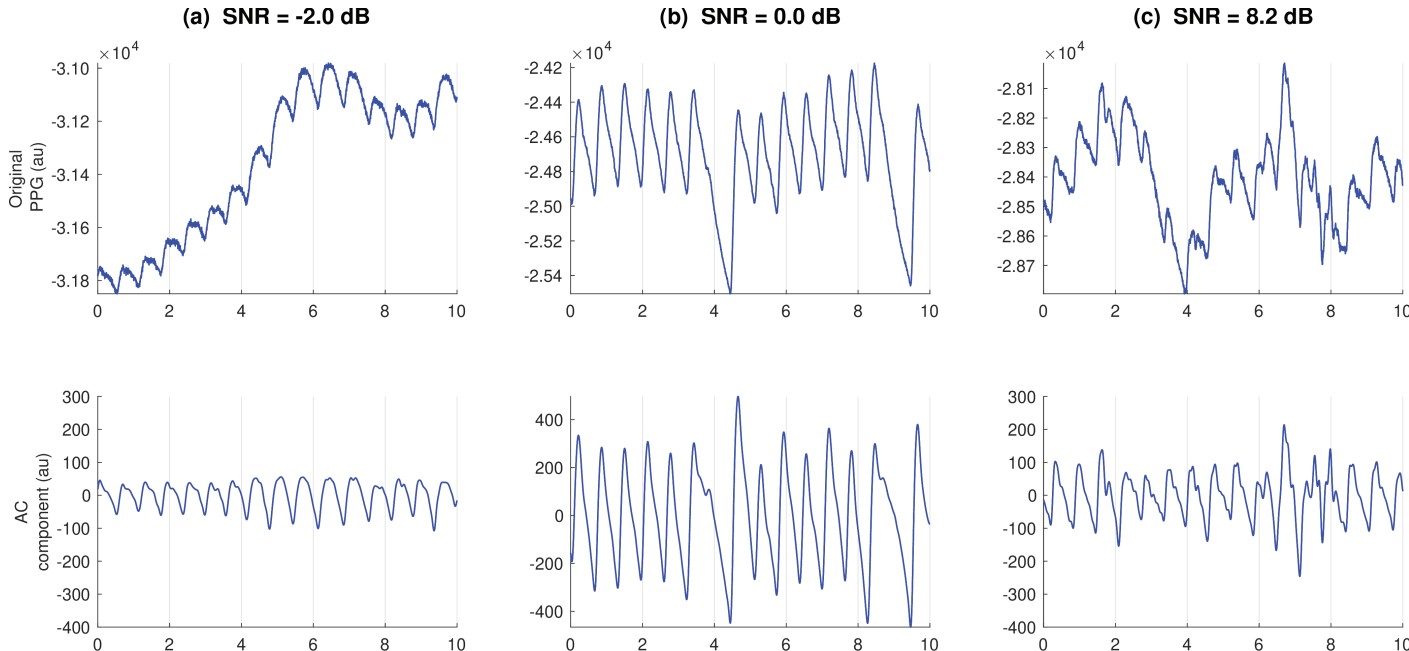

**Fig 4. Examples of PPG signals whose SNR values did not align with their quality**: (**a**) a reasonably high-quality PPG signal (individual pulse waves are clearly visible and details of their shapes are also visible) with a low SNR value due to a noisy frequency spectrum; (**b**) a high-quality PPG signal with a low SNR value due to the nonstationary nature of the signal produced by an irregular rhythm (e.g., premature ventricular contractions); and (**c**) a medium-quality PPG signal (individual pulse waves are visible but their shapes are corrupted by noise) and yet a reasonably high SNR value.

15.5 (11.3–19.2) dB at heart height, vs. 13.7 (8.2–17.8) dB when held in lap, vs. 10.5 (5.4–15.2) dB when hanging down; all p < 0.001).

### Influence of participant characteristics on PPG signal quality

Tables 5, 6 and 7 show the results of mixed effects analyses of the influences of participant characteristics on PPG signal quality when assessing signal quality using the SNR, PI, and TMCC respectively. In each table, results are presented for both the Auscultatory and Oscillometric Subsets. The findings, which were mixed across the two subsets and across different quality metrics, are now described.

Signal quality increased with age when assessed using metrics indicative of the periodicity of the signal (the SNR and TMCC). On the other hand, age was not significantly associated with the PI. Gender was significantly associated with the TMCC on both subsets (with higher signal quality in males), but not with the SNR or TMCC. BMI was significantly associated with the SNR and TMCC on the Auscultatory Subset, but was not associated with any quality metrics on the Oscillometric Subset. There were no associations between diabetes and quality metrics. Skin tone was significantly associated with the SNR on the Oscillometric subset (where darker skin types were associated with lower quality), and darker skin types were associated with higher PIs on the Auscultatory Subset. The impact of blood pressure on signal quality varied: increasing systolic blood pressure was associated with higher SNR and PI on the Auscultatory Subset but not the Oscillometric Subset, and with lower TMCC on both subsets. Increasing pulse pressure was associated with lower SNR and PI on the Auscultatory Subset, but with higher TMCC on both subsets, and higher PI on the Oscillometric Subset.

## Comparison between postures (with sensor arm in natural position)

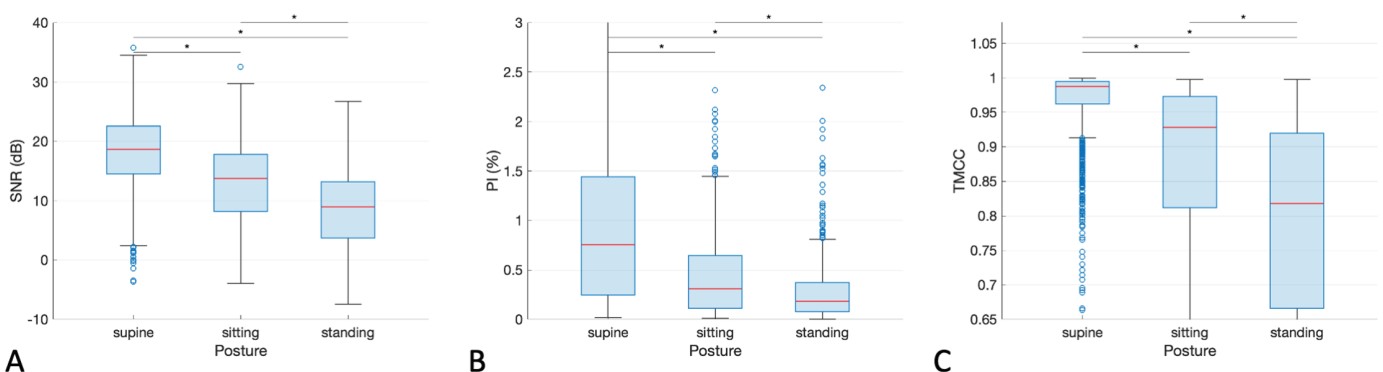

## Comparison between postures (with sensor at heart height)

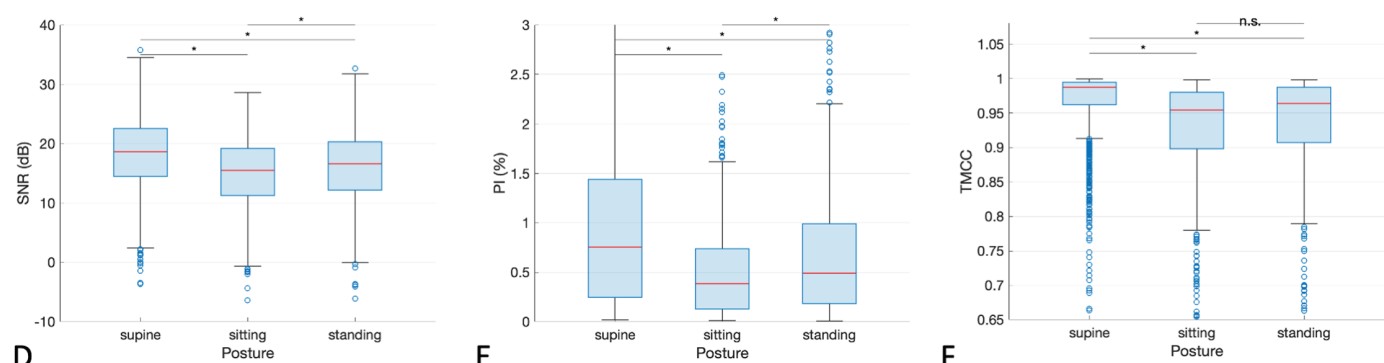

**Fig 5. The quality of PPG signals measured in different postures**: (upper row, **A–C**) comparison between postures with arm in the natural position for each posture; (lower row, **D–F**) comparison between postures with sensor at heart height. Each row shows: (left, A and D) the SNR; (middle, B and E) the PI; and (right, C and F) the TMCC. *Definition: An asterisk (\*) indicates a statistically significant difference.*

Higher DC amplitudes were associated with lower SNRs and PIs on both subsets, and with higher TMCCs on the Auscultatory Subset only.

Table 8 presents the results obtained through a second mixed effects analysis of the influences on the SNR, this time not accounting for DC amplitude. This analysis revealed that when DC amplitude was not accounted for, darker skin tones were significantly associated with higher quality on the Auscultatory Subset, but no association was observed on the Osillometric Subset. In contrast, when DC amplitude was included (Table 5), darker skin tones were associated with lower signal quality on the Osillometric Subset, but no association was observed on the Auscultatory Subset.

Fig 7 demonstrates the observed relationship between the SNR and its key determinants. The upper row corresponding to the Auscultatory Subset shows that, when holding all other variables constant (and including DC amplitude in the model):

- the SNR would be expected to increase from 16 to 24 dB as participant age increases from 21 to 85, an increase of 1.2 dB per decade;
- the SNR would be expected to decrease from 20 to 18 dB as skin tone varied from Fitzpatrick Type 1 (palest) to VI (darkest), a reduction of 0.5 dB per unit increment on the Fitzpatrick scale;

## Comparison between sensor heights (whilst standing)

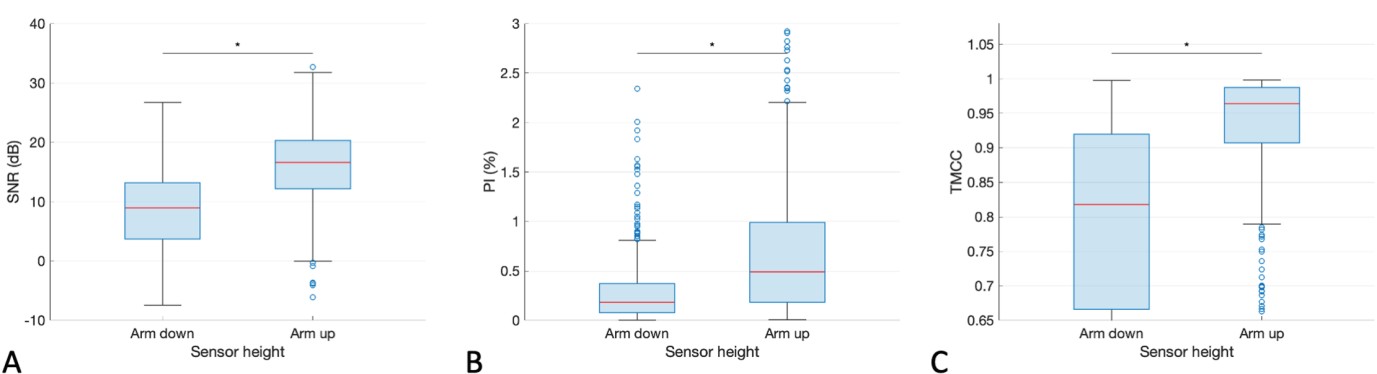

## Comparison between sensor heights (whilst sitting)

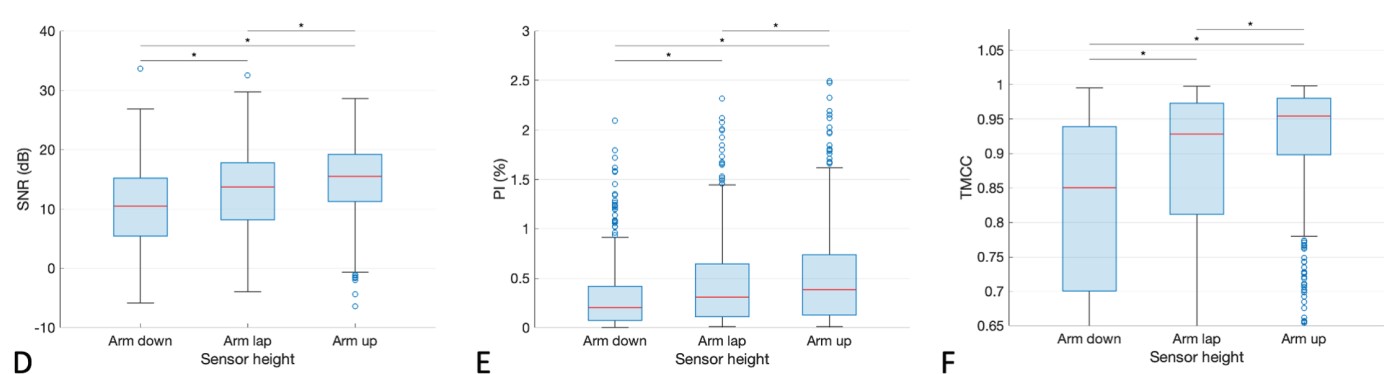

**Fig 6. The quality of PPG signals measured at different sensor heights.** Results are shown for: (upper row, **A–C**) standing; and (lower row, **D–F**) sitting. Each row shows: (left, A and D) the SNR; (middle, B and E) the PI; and (right, C and F) the TMCC. *Definition: An asterisk (\*) indicates a statistically significant difference.*

- the SNR would be expected to decrease from 22 to 14 dB as DC amplitude increases from the lowest to highest value in the subset (from 3k to 875k).;

In addition, the lower row corresponding to the Oscillometric Subset demonstrates the influences of sensor height and posture. As also found in the paired data analysis, sensor height had the greatest impact on signal quality, with the SNR increasing by ≈2.9 dB when elevating the sensor from an 'arm down' position to 'arm in lap' position, and by ≈3.3 dB from and 'arm in lap' position to 'arm up' position. The signal quality was also increased in the supine position compared to sitting by ≈2.7 dB.

## Discussion

### Summary of findings

This study investigated the impact of several factors on the quality of wrist PPG signals. Posture and sensor height relative to the heart were identified as key determinants of PPG signal quality. Signal quality was highest in the supine posture, and lower when sitting or standing. Signal quality was highest when the sensor was at heart height, and lower at lower

**Table 5. The influence of participant characteristics on PPG signal quality, assessed using the SNR.**

| Characteristic | Coefficient, β | | Statistics | |
|---|---|---|---|---|
| | Estimate (95% CIs) | Standard Error | t-statistic | p-value |
| *Auscultatory Subset* | | | | |
| Age (years) | 0.12 (0.08–0.17) | 0.02 | 5.33 | 0.000* |
| Gender (F = 1, M = 2) | 0.15 (–1.00 to 1.30) | 0.58 | 0.26 | 0.796 |
| BMI (kgm$^{-3}$) | –0.10 (–0.19 to –0.01) | 0.05 | –2.08 | 0.038* |
| Diabetes (present = 1, absent = 0) | –0.90 (–3.01 to 1.21) | 1.08 | –0.84 | 0.403 |
| Skin color (Fitzpatrick skin type, 1–6) | –0.52 (–1.20 to 0.16) | 0.35 | –1.49 | 0.137 |
| Systolic blood pressure (mmHg) | 0.07 (0.03 to 0.11) | 0.02 | 3.47 | 0.001* |
| Pulse pressure (mmHg) | –0.15 (–0.19 to –0.10) | 0.02 | –6.44 | 0.000* |
| DC amplitude (arbitrary units) | -0.000009 (–0.000012 to –0.000006) | 0.000001 | –6.30 | 0.000* |
| *Oscillometric Subset* | | | | |
| Age (years) | 0.17 (0.09 to 0.26) | 0.04 | 4.04 | 0.000* |
| Gender (F = 1, M = 2) | 0.77 (–0.83 to 2.37) | 0.82 | 0.94 | 0.346 |
| BMI (kgm$^{-3}$) | 0.00 (–0.10 to 0.11) | 0.05 | 0.06 | 0.953 |
| Diabetes (present = 1, absent = 0) | –1.26 (–4.03 to 1.51) | 1.41 | –0.89 | 0.373 |
| Skin color (Fitzpatrick skin type, 1–6) | –0.81 (–1.61 to –0.02) | 0.40 | –2.01 | 0.045* |
| Systolic blood pressure (mmHg) | 0.01 (–0.05 to 0.06) | 0.03 | 0.23 | 0.819 |
| Pulse pressure (mmHg) | 0.04 (–0.03 to 0.11) | 0.03 | 1.18 | 0.239 |
| DC amplitude (arbitrary units) | –0.000008 (–0.000011 to –0.000004) | 0.000002 | –3.89 | 0.000* |
| Sitting posture (vs. standing) | 0.53 (–0.25 to 1.31) | 0.40 | 1.34 | 0.181 |
| Supine posture (vs. standing) | 2.90 (1.89 to 3.92) | 0.52 | 5.61 | 0.000* |
| Arm up (vs. arm down) | 6.67 (5.88 to 7.45) | 0.40 | 16.62 | 0.000* |
| Hand in lap (vs. arm down) | 3.23 (2.20 to 4.27) | 0.53 | 6.12 | 0.000* |

**Table 6. The influence of participant characteristics on PPG signal quality, assessed using the TMCC.**

| Characteristic | Coefficient, β | | Statistics | |
|---|---|---|---|---|
| | Estimate (95% CIs) | Standard Error | t-statistic | p-value |
| *Auscultatory Subset* | | | | |
| Age (years) | 0.001 (0.000–0.001) | 0.000 | 2.95 | 0.003* |
| Gender (F = 1, M = 2) | 0.015 (0.002–0.029) | 0.007 | 2.20 | 0.028* |
| BMI (kgm$^{-3}$) | 0.001 (0.000–0.002) | 0.001 | 2.28 | 0.023* |
| Diabetes (present = 1, absent = 0) | –0.011 (–0.036–0.015) | 0.013 | –0.83 | 0.408 |
| Skin color (Fitzpatrick skin type, 1–6) | –0.003 (–0.011–0.005) | 0.004 | –0.67 | 0.502 |
| Systolic blood pressure (mmHg) | –0.001 (–0.002 to –0.001) | 0.000 | –5.93 | 0.000* |
| Pulse pressure (mmHg) | 0.001 (0.000–0.001) | 0.000 | 2.73 | 0.006* |
| DC amplitude (arbitrary units) | 0.000000 (0.000000–0.000000) | 0.000000 | 2.93 | 0.003* |
| *Oscillometric subset* | | | | |
| Age (years) | 0.003 (0.001–0.005) | 0.001 | 3.22 | 0.001* |
| Gender (F = 1, M = 2) | 0.038 (0.004–0.072) | 0.017 | 2.21 | 0.028* |
| BMI (kgm$^{-3}$) | 0.001 (–0.001–0.003) | 0.001 | 0.91 | 0.361 |
| Diabetes (present = 1, absent = 0) | –0.056 (–0.115–0.003) | 0.030 | –1.85 | 0.064 |
| Skin color (Fitzpatrick skin type, 1–6) | –0.011 (–0.028–0.006) | 0.009 | –1.30 | 0.194 |
| Systolic blood pressure (mmHg) | –0.002 (–0.003 to –0.001) | 0.001 | –3.92 | 0.000* |
| Pulse pressure (mmHg) | 0.003 (0.001–0.004) | 0.001 | 4.10 | 0.000* |
| DC amplitude (arbitrary units) | 0.000000 (–0.000000–0.000000) | 0.000000 | 0.72 | 0.469 |
| Sitting posture (vs. standing) | 0.009 (–0.007–0.025) | 0.008 | 1.07 | 0.285 |
| Supine posture (vs. standing) | 0.023 (0.002–0.044) | 0.011 | 2.18 | 0.029* |
| Arm up (vs. arm down) | 0.133 (0.117–0.149) | 0.008 | 16.12 | 0.000* |
| Hand in lap (vs. arm down) | 0.079 (0.057–0.100) | 0.011 | 7.25 | 0.000* |

elevations. In addition, findings were mixed on the potential influences on signal quality of age, skin tone, blood pressure, and DC amplitude. We now present potential explanations for the observed associations with signal quality.

**Table 7. The influence of participant characteristics on PPG signal quality, assessed using the PI.**

| Characteristic | Coefficient, β Estimate (95% CIs) | Standard Error | Statistics t-statistic | p-value |
|---|---|---|---|---|
| *Auscultatory Subset* | | | | |
| Age (years) | −0.00 (−0.00–0.00) | 0.00 | −0.67 | 0.504 |
| Gender (F = 1, M = 2) | 0.02 (−0.03–0.06) | 0.02 | 0.76 | 0.445 |
| BMI (kgm$^{-3}$) | 0.00 (−0.00–0.00) | 0.00 | 0.52 | 0.606 |
| Diabetes (present = 1, absent = 0) | 0.03 (−0.06–0.11) | 0.04 | 0.62 | 0.535 |
| Skin colour (Fitzpatrick skin type, 1–6) | 0.09 (0.07–0.12) | 0.01 | 7.57 | 0.000* |
| Systolic blood pressure (mmHg) | 0.00 (0.00–0.00) | 0.00 | 3.22 | 0.001* |
| Pulse pressure (mmHg) | -0.00 (−0.00 to −0.00) | 0.00 | −2.32 | 0.021* |
| DC amplitude (arbitrary units) | −0.000001 (−0.000001 to −0.000001) | 0.000000 | −15.72 | 0.000* |
| *Oscillometric Subset* | | | | |
| Age (years) | 0.00 (−0.00–0.01) | 0.00 | 0.43 | 0.665 |
| Gender (F = 1, M = 2) | 0.10 (−0.02–0.22) | 0.06 | 1.61 | 0.108 |
| BMI (kgm$^{-3}$) | 0.00 (−0.01–0.01) | 0.00 | 0.46 | 0.644 |
| Diabetes (present = 1, absent = 0) | −0.19 (−0.40–0.02) | 0.11 | −1.78 | 0.075 |
| Skin color (Fitzpatrick skin type, 1–6) | 0.02 (−0.04–0.08) | 0.03 | 0.71 | 0.476 |
| Systolic blood pressure (mmHg) | -0.00 (−0.01–0.00) | 0.00 | −1.60 | 0.109 |
| Pulse pressure (mmHg) | 0.01 (0.00–0.01) | 0.00 | 2.83 | 0.005* |
| DC amplitude (arbitrary units) | −0.000002 (−0.000002 to −0.000001) | 0.000000 | −12.36 | 0.000* |
| Sitting posture (vs. standing) | −0.03 (−0.08–0.01) | 0.02 | −1.48 | 0.139 |
| Supine posture (vs. standing) | 0.23 (0.17–0.29) | 0.03 | 7.58 | 0.000* |
| Arm up (vs. arm down) | 0.25 (0.21–0.30) | 0.02 | 10.91 | 0.000* |
| Hand in lap (vs. arm down) | 0.16 (0.10–0.22) | 0.03 | 5.25 | 0.000* |

**Table 8. The influence of participant characteristics on PPG signal quality (without including DC amplitude), assessed using the SNR.**

| Characteristic | Coefficient, β Estimate (95% CIs) | Standard Error | Statistics t-statistic | p-value |
|---|---|---|---|---|
| *Auscultatory Subset* | | | | |
| Age (years) | 0.12 (0.08–0.17) | 0.02 | 5.13 | 0.000* |
| Gender (F = 1, M = 2) | 0.55 (−0.61–1.71) | 0.59 | 0.93 | 0.353 |
| BMI (kgm$^{-3}$) | −0.09 (−0.19–0.00) | 0.05 | −1.91 | 0.057 |
| Diabetes (present = 1, absent = 0) | −1.36 (−3.50–0.78) | 1.09 | −1.25 | 0.213 |
| Skin color (Fitzpatrick skin type, 1–6) | 0.82 (0.28–1.37) | 0.28 | 2.95 | 0.003* |
| Systolic blood pressure (mmHg) | 0.08 (0.04–0.12) | 0.02 | 3.78 | 0.000* |
| Pulse pressure (mmHg) | -0.16 (−0.20 to −0.11) | 0.02 | −6.87 | 0.000* |
| *Oscillometric Subset* | | | | |
| Age (years) | 0.14 (0.05–0.22) | 0.04 | 3.03 | 0.003* |
| Gender (F = 1, M = 2) | 1.32 (−0.31–2.96) | 0.83 | 1.59 | 0.113 |
| BMI (kgm$^{-3}$) | 0.01 (−0.10–0.12) | 0.06 | 0.20 | 0.845 |
| Diabetes (present = 1, absent = 0) | −0.86 (−3.73–2.02) | 1.47 | −0.59 | 0.558 |
| Skin color (Fitzpatrick skin type, 1–6) | −0.10 (−0.86–0.66) | 0.39 | −0.26 | 0.797 |
| Systolic blood pressure (mmHg) | 0.03 (−0.03–0.08) | 0.03 | 0.97 | 0.333 |
| Pulse pressure (mmHg) | 0.02 (−0.05–0.09) | 0.03 | 0.53 | 0.596 |
| Sitting posture (vs. standing) | 0.52 (−0.29–1.32) | 0.41 | 1.26 | 0.207 |
| Supine posture (vs. standing) | 3.23 (2.19–4.28) | 0.53 | 6.06 | 0.000* |
| Arm up (vs. arm down) | 6.05 (5.24–6.86) | 0.41 | 14.73 | 0.000* |
| Hand in lap (vs. arm down) | 2.90 (1.83–3.97) | 0.54 | 5.33 | 0.000* |

The association of posture (vertical versus supine) with PPG signal quality may be due to regulatory cardiovascular mechanisms, e.g., redistribution of blood, changes in stroke volume and dynamic vascular stiffness. In the vertical position, gravity causes blood to pool in

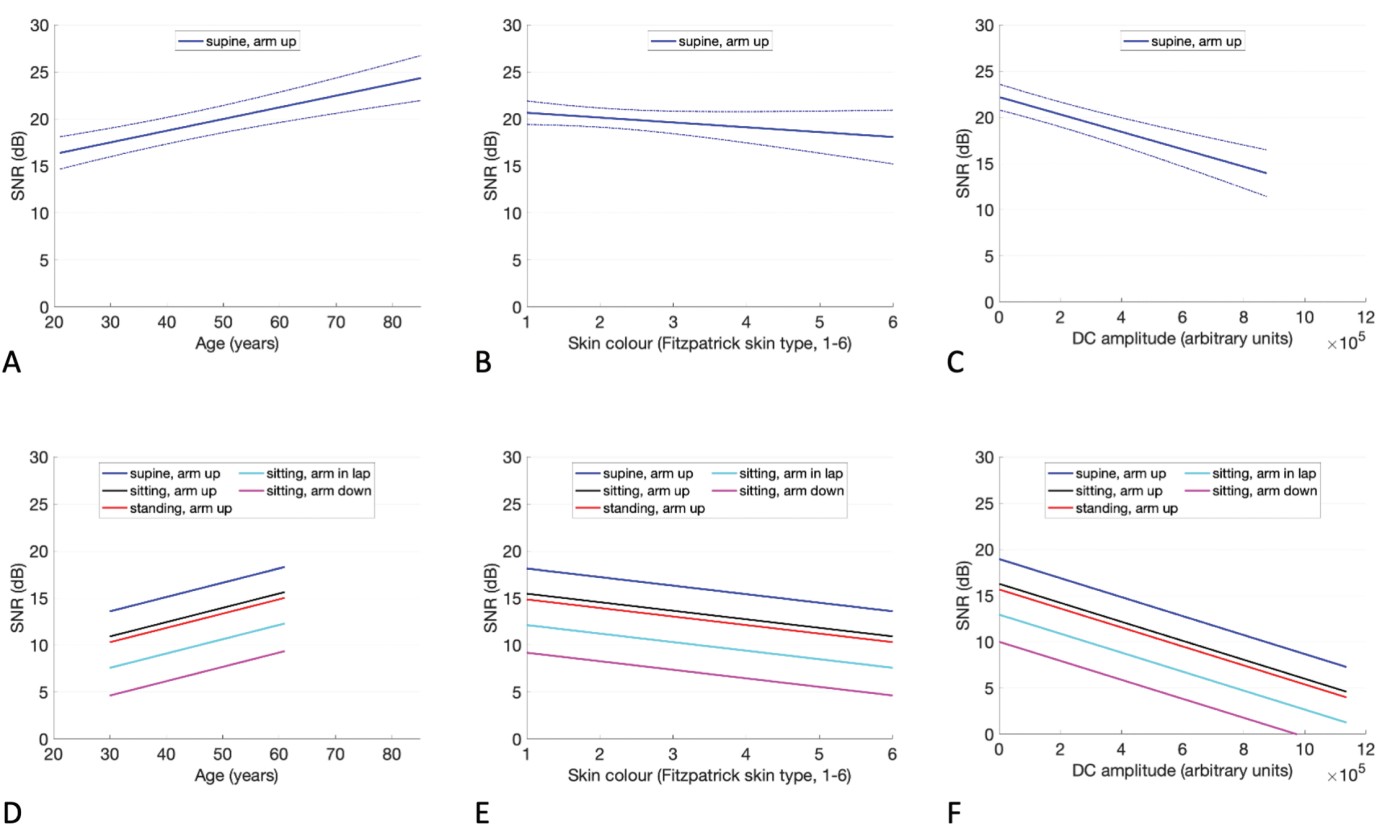

**Fig 7. Determinants of the PPG SNR:** The plots show the modelled relationship between the SNR and its key determinants on: (upper row, **A–C**) the Auscultatory Subset; and (lower row, **D–F**) the Oscillometric Subset. Solid lines indicate predicted values when all other parameters were set to normal values. In the upper row dashed lines indicate non-simultaneous 95% confidence intervals.

the lower extremities, reducing venous return to the heart and decreasing end-diastolic volume (preload), stroke volume, and cardiac output [62,63]. To compensate, the body increases heart rate via the baroreceptor reflex and induces vasoconstriction through sympathetic nervous system activity to maintain blood pressure. Conversely, lying down eliminates the effect of gravity on blood pooling, increasing venous return and enhancing preload, stroke volume, and cardiac output via the Frank-Starling mechanism. As a result, the heart rate decreases, and the body adjusts through vasodilation and decreased sympathetic activity to stabilize blood pressure. All the above cardiovascular changes directly affect PPG signal quality. For example, in the vertical position, venous pooling increases incident light absorption, while diminished stroke volume and increased vasoconstriction decrease blood volume pulsations and PPG amplitude.

Similarly, PPG sensor height affects PPG signal quality mainly due to venous blood redistribution, pooling in the lowered limb, increased drainage in the heightened limb, and transmural pressure changes. When a subject's hand with a PPG sensor on the wrist is in a normal position, i.e., to the side below heart level, venous blood flow has to overcome a greater hydrostatic pressure difference, which is approximately +35 mmHg, to return to the heart [64,65]. Higher venous blood volumes result in increased light absorption by non-pulsatile blood [18], thus reducing signal quality. When the limb is heightened, or at heart level, a negative hydrostatic pressure difference (-10 to 0 mmHg) helps to drain venous blood and thus decreases

light absorption. When lying supine, the effects of hydrostatic pressure are abolished, and regional differences in venous pressure are relatively small. In addition, PPG signal amplitude changes are due to the transmural pressure difference between pressures inside and outside the arterial walls [66]. It is known that arterial compliance is maximal when the transmural pressure is zero, i.e., when arterial walls are at equilibrium, the PPG amplitude is maximal. The results in [66] indicate that the zero-point was reached between 60-80 cm above heart level.

The observed associations between age and signal quality metrics indicative of periodicity (SNR and TMCC) are probably due to decreased heart rate variability and decreased complexity of the pulse wave shape at older ages. Heart rate variability reduces with age [67], which will reduce pulse rate variability in the PPG, resulting in more regular PPG signals in older subjects. In addition, the PPG's secondary diastolic peak and dicrotic notch disappear with increasing age, resulting in a less complex PPG signal which is closer to sinusoidal in shape. Both the reduced variability and reduced complexity of PPG pulse wave shape at older ages will result in increased SNR and TMCC, even though the 'quality' of the measurement (i.e., how faithfully it captures true blood flow patterns) may not differ.

The results of this study do not provide clear indication of how skin tone or blood pressure relate to signal quality. The reduction in SNR in darker skin tones observed on the Oscillometric subset (where the p-value was only slightly below the significance threshold) could be explained by increased light absorbance in the epidermis of darker skin due to increased melanin levels, resulting in less light being reflected back to the sensor [68]. However, there was no corresponding reduction in SNR on the Auscultatory Subset, nor in TMCC in darker skin tones, and the PI increased in darker skin tones on the Auscultatory Subset, creating uncertainty in the interpretation of this result. The comparison of models including and not including DC amplitude indicates that in these data the association between skin tone and signal quality (via the SNR) was moderated by the DC amplitude: when not accounting for DC amplitude, darker skin types were associated with higher quality on the Auscultatory Subset, and there was no significant association between skin tone and signal quality on the Oscillometric Subset, illustrating real-world observations. This illustrates the potential benefits of adjusting LED light intensity to maintain signal quality across different skin tones: on the Oscillometric Subset, the reduction in signal quality associated with darker skin tones when controlling for LED light intensity (*i.e. DC amplitude*) was no longer present in the real-world scenario of varying light intensity to account for skin tone. Similarly, on the Auscultatory Subset, the signal quality became significantly higher at darker skin tones when varying light intensity. Inconsistent relationships were observed between blood pressure and signal quality. It should be noted that several potential confounders may influence the observed relationships for these variables, such as: (i) LED light intensity; (ii) LED geometry; (iii) source-detector distance; and (iv) the wavelength(s) of LED light (see Table 1 for details of these factors). Each of these could influence the region of vasculature from which the PPG signal is obtained, thus influencing the physiological origins of the signal.

## Comparison with literature

This study adds to our understanding of factors that influence PPG signal quality. One factor which has received relatively little attention in the literature is the height of the PPG sensor relative to the heart. It is reasonable to expect that changes in the height of a PPG measurement site (such as the wrist being raised from the arm hanging by one's side and to being elevated) may influence signal quality for multiple reasons. First, the arterial blood pressure at the site will change due to changes in hydrostatic pressure (i.e., the gravitational component of

blood pressure drops as the arm is raised). Second, the volume of blood (arterial and venous) at the measurement site may change (i.e., the volume reduces as the arm is raised). Indeed, it has previously been observed that sensor height impacts the PPG signal: the amplitude of the pulsatile (AC) component of the PPG signal was found to decrease at higher measurement sites in [69] (corresponding to lower signal quality), whilst it was found to increase at higher measurement sites in [16,17] (corresponding to higher signal quality). Both AC and DC components were found to increase at higher measurement sites in [18]. Smaller changes were observed in [19]. These studies were conducted in relatively low numbers of participants (6–20), with the PPG signal measured at either the finger [16–19] or toe [69] rather than the wrist which is commonly used by smartwatches and fitness trackers [3]. Consequently, prior to this study it was not clear how the height of the PPG measurement site relative to the heart influences signal quality, particularly in the case of wrist PPG signals. In this study the PPG signal quality consistently increased at higher sensor heights, regardless of which signal quality metric was used.

Previous research has observed signal quality to be lower during the day than whilst sleeping at night [45–47], and some PPG-derived measurements such as resting heart rate are sometimes specifically measured at night [70]. Several factors may contribute to improved signal quality at night, including lower levels of movement and lower ambient light levels. The present study indicates that posture and sensor height may also contribute to this, with the supine posture typically adopted whilst sleeping with the sensor at approximately heart height resulting in higher signal quality than sitting and standing postures during the day, where the sensor is usually below heart height. In this study signal quality was consistently higher in the supine position, regardless of which signal quality metric was used. Whilst PPG signals measured during light activities (such as walking) are not often subject to detailed analysis because of motion artifact, this study highlights an additional challenge to using such data: that signal quality is lower in the standing position, particularly when the sensor is below heart height.

## Strengths and limitations

A first strength of this study is that it is based on analysis of a large dataset of PPG signals. The signals were collected from 1,142 subjects with a range of health statuses and skin colors, in a variety of postures and with controlled changes in hand height. This allowed us to investigate the influence of several factors on signal quality. Second, signal quality was assessed using several techniques, allowing us to investigate whether findings were consistent across different techniques. Third, the study used reflectance PPG signals measured at the wrist, which are representative of those measured by commonly-used smartwatches and fitness trackers.

A key limitation of this study is that the findings are based on analysis of green wavelength PPG signals. Therefore, it is not clear how generalizable the findings would be to other wavelengths of light. Future work should assess how the quality of PPG signals differs when using different wavelengths of light, particularly with subjects with varying skin tones. More generally, the PPG signals were collected using one particular device attached at a particular anatomical site, and it is not yet clear how generalizable the findings are to other wearable photoplethysmography devices, which could be attached at different sites. An additional limitation is that we have not attempted to investigate how the observed differences in PPG signal quality affect PPG analysis. It is likely that different levels of signal quality will be required for different analysis tasks: it may be possible to estimate heart rate from relatively low-quality signals (such as the low- and medium-quality signals shown in Fig 1), whereas higher-quality

signals may be required for analyses of PPG pulse wave shape (such as blood pressure estimation and vascular aging assessment).

## Implications

This study provides insights into how to optimize PPG signal quality. PPG signal quality was found to be improved when in the supine position, and when the sensor was positioned at (rather than below) heart height. This adds to the body of evidence indicating that PPG signals obtained during sleep (corresponding to the supine position) may be more suitable for analysis than those obtained during other activities of daily living. This is particularly relevant with the emergence of novel PPG parameters obtained during sleep, such as nocturnal pulse wave amplitude attenuations for cardiovascular risk assessment [71]. In addition, the findings support mounting PPG sensors on body parts that exhibit smaller hydrostatic pressure differences relative to heart level (e.g., upper arm, chest) [72]. When using wrist devices, it may be advantageous to incorporate automated methods to determine whether PPG signals are likely to be suitable for analysis. For instance, simultaneous accelerometry could be used to identify when the arm is hanging by the side of the body (derived from the sensor's orientation), indicating that the sensor is substantially below heart height and signal quality is expected to be lower. Even when the subject is optimally positioned for high-quality PPG recording (supine or with sensor close to heart height), it should be noted that the PPG is highly susceptible to artifacts and distortions, meaning additional methods will be required to monitor and control for confounding factors such as contact pressure.

This study also highlights key areas for future research. First, it is important to investigate further the impacts of participant characteristics (age, skin tone, and blood pressure) on PPG signal quality. Differences in PPG signal quality associated with physiology could inform physiological assessment, such as whether signal quality is increased in hypertension, or decreased in low-pulse pressure scenarios (such as peripheral vasoconstriction, potentially associated with clinical deterioration). Furthermore, any differences in signal quality associated with skin tone may be more pronounced with different light wavelengths, highlighting the importance of identifying optimal light wavelengths for use with different skin tones. Second, it is important to investigate the impacts of device characteristics (e.g., LED light intensity) on PPG signal quality, and to determine optimal strategies for automatically adjusting light intensity to optimise signal quality, particularly across different skin tones. Here, there is a compromise between increasing light intensity to increase signal quality, and decreasing light intensity to reduce power consumption and local heating effects. Third, it is clear from this study that traditional metrics of PPG signal quality are far from perfect (e.g., see Fig 4), and this adds importance to research into PPG signal quality assessment, particularly techniques which quantify quality numerically rather than producing a classification.

The analysis code used in this study is openly available at https://github.com/peterhcharlton/PPG_quality_determinants [73], and may be useful to future researchers using the Aurora-BP dataset.

## Conclusion

In this study of PPG signals acquired using a wrist-worn reflectance photoplethysmography device, signal quality was found to be associated with posture and the height of the sensor relative to the heart. Signal quality was highest in the supine posture, and when the sensor was at heart height. Signal quality was greatly reduced in sitting and standing postures, particularly when the sensor was held in natural positions rather than at heart height. Signal quality was either reduced or constant at darker skin tones when controlling for the DC amplitude of the

PPG signal, whereas it was either constant or increased at darker skin tones when allowing the DC amplitude to vary, highlighting the potential benefit of adjusting LED light intensity to maintain signal quality across different skin tones. Future work should consider how to optimise PPG signal acquisition, how to identify periods of PPG signal quality which are of sufficient quality for analysis, and how participant and device characteristics may also influence PPG signal quality. Ultimately, such work is expected to lead to robust processes for obtaining high-quality PPG signals in daily life.

## Acknowledgments

ChatGPT (OpenAI, San Francisco, CA, USA) was used for language editing.

## Author contributions

**Conceptualization:** Peter H Charlton, Panicos A. Kyriacou.

**Data curation:** Peter H Charlton.

**Formal analysis:** Peter H Charlton.

**Funding acquisition:** Peter H Charlton, Panicos A. Kyriacou, Jonathan Mant.

**Investigation:** Peter H Charlton, Jonathan Mant.

**Methodology:** Peter H Charlton, Vaidotas Marozas, Elisa Mejía-Mejía, Panicos A. Kyriacou, Jonathan Mant.

**Project administration:** Peter H Charlton, Panicos A. Kyriacou, Jonathan Mant.

**Resources:** Peter H Charlton.

**Software:** Peter H Charlton.

**Supervision:** Panicos A. Kyriacou, Jonathan Mant.

**Visualization:** Peter H Charlton, Vaidotas Marozas.

**Writing – original draft:** Peter H Charlton, Vaidotas Marozas.

**Writing – review & editing:** Peter H Charlton, Vaidotas Marozas, Elisa Mejía-Mejía, Panicos A. Kyriacou, Jonathan Mant.

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
