## [Decision Letter · Decision Letter 0]

22 Oct 2024

PDIG-D-24-00286

Determinants of photoplethysmography signal quality at the wrist

PLOS Digital Health

Dear Dr. Charlton,

Thank you for submitting your manuscript to PLOS Digital Health. After careful consideration, we feel that it has merit but does not fully meet PLOS Digital Health's publication criteria as it currently stands. Therefore, we invite you to submit a revised version of the manuscript that addresses the points raised during the review process.

Please submit your revised manuscript within 60 days Dec 21 2024 11:59PM. If you will need more time than this to complete your revisions, please reply to this message or contact the journal office at digitalhealth@plos.org. Please include the following items when submitting your revised manuscript:

We look forward to receiving your revised manuscript.

Kind regards,

Walter Karlen

Academic Editor

PLOS Digital Health

Walter Karlen

Academic Editor

PLOS Digital Health

Journal Requirements:

1. We ask that a manuscript source file is provided at Revision. Please upload your manuscript file as a .doc, .docx, .rtf or .tex.

Additional Editor Comments (if provided):

Dear authors, the reviewers did find some irregularities regarding the reported statistics, please verify your employed statistical methods and correct where necessary or justify your chosen approach. Please also consider the suggestions regarding the structure and legibility of the manuscript.

Reviewers' comments:

Reviewer's Responses to Questions

**Comments to the Author**

1. Does this manuscript meet PLOS Digital Health’s publication criteria? Is the manuscript technically sound, and do the data support the conclusions? The manuscript must describe methodologically and ethically rigorous research with conclusions that are appropriately drawn based on the data presented.

Reviewer #1: Yes

Reviewer #2: Yes

2. Has the statistical analysis been performed appropriately and rigorously?

Reviewer #1: Yes

Reviewer #2: Yes

3. Have the authors made all data underlying the findings in their manuscript fully available (please refer to the Data Availability Statement at the start of the manuscript PDF file)?

Reviewer #1: Yes

Reviewer #2: Yes

4. Is the manuscript presented in an intelligible fashion and written in standard English?

PLOS Digital Health does not copyedit accepted manuscripts, so the language in submitted articles must be clear, correct, and unambiguous. Any typographical or grammatical errors should be corrected at revision, so please note any specific errors here.

Reviewer #1: Yes

Reviewer #2: Yes

5. Review Comments to the Author

Please use the space provided to explain your answers to the questions above. You may also include additional comments for the author, including concerns about dual publication, research ethics, or publication ethics. (Please upload your review as an attachment if it exceeds 20,000 characters)

Reviewer #1: The article outlines a methodology for analyzing the key characteristics that ensure high-quality photoplethysmogram (PPG) signal acquisition. While many aspects related to sensor performance have been extensively studied, the impact of factors such as skin tone, sensor height relative to the heart, and posture remains insufficiently explored. The authors aim to quantify the influence of these variables in their study. The article is well-structured and presented in a clear, accessible format. However, I do have some suggestions to improve the manuscript a bit:

1.The introduction lacks sufficient detail on existing literature in this field. While the authors attempt to contextualize their findings within the broader body of research in the discussions section later, it would be beneficial to present these works earlier in the introduction and revisit them in the discussion for better perspective.

2. Please provide the equation used to calculatedSNR.

3. Regarding the correlation with a template, was the template derived from a clean segment of the signal, or was it averaged over the entire signal record? If the template is also generated from noisy data, the template may also be noisy, leading to concerns on whether correlation with such a template would be an indicator of signal quality. Please discuss this a bit more in the method.

4. In the mixed-effects analysis, the process by which subject-specific intercepts (random effects) were obtained is unclear. Please clarify this aspect of the methodology.

I look forward to reading your article after the revisions.

Reviewer #2: The authors present an extensive analysis on factors influencing wrist PPG signal quality. The work uses three signal quality measures to evaluate the influence of posture, sensor height and participant characteristics on a large freely available dataset. Overall, the work is well written and important fundamental research.

I have some minor comments and questions for the authors:

- In Figure 1 no unit on the y axis is shown. As in Figure 2, the unit should be set to a.u..

- In line 143 it is stated that the Wilcoxon tests were performed at a significance level of 5%. In figure 5 and 6, statistically significant differences are reported, when p<0.001, indicating a significance level of 0.1%. The authors should unify this.

- Also, in Figure 5 and 6, there are some indicators of significance in brackets, others are not. Does this have any meaning? If so, the authors should state this in the caption. Also, in Figure 6, the statement regarding significant differences in the lower row on the left needs more space. Generally, the authors should consider using significance bars in the plot for easier to grasp significant differences.

- If I understand correctly, each group in the Wilcoxon test comparisons contains multiple observations of the same subjects. If this is the case, then in such a comparison, the subjects should be regarded as a random effect, because the samples of a group are not independent anymore.

- Also in the conducted statistical analysis (Figure 5 and 6), the authors compare all groups against one another. Therefore, the analysis should be corrected for multiple comparisons.

- Two minor spelling mistakes caught my eye:

o In line 56 'acqusition' should be changed to 'acquisition'.

o In line 249 'event' should be changed to 'even'.

6. PLOS authors have the option to publish the peer review history of their article (what does this mean?). If published, this will include your full peer review and any attached files.

**Do you want your identity to be public for this peer review?** For information about this choice, including consent withdrawal, please see our Privacy Policy.

Reviewer #1: No

Reviewer #2: No

---

## [Decision Letter · Decision Letter 1]

9 Apr 2025

Determinants of photoplethysmography signal quality at the wrist

PDIG-D-24-00286R1

Dear Dr Charlton,

We are pleased to inform you that your manuscript 'Determinants of photoplethysmography signal quality at the wrist' has been provisionally accepted for publication in PLOS Digital Health.

Best regards,

Walter Karlen

Academic Editor

PLOS Digital Health

**Additional Editor Comments (if provided):**

Dear Authors

Thank you for your thorough revision, which was to the satisfaction of the reviewer

**Reviewer Comments (if any, and for reference):**

Reviewer's Responses to Questions

**Comments to the Author**

1. If the authors have adequately addressed your comments raised in a previous round of review and you feel that this manuscript is now acceptable for publication, you may indicate that here to bypass the “Comments to the Author” section, enter your conflict of interest statement in the “Confidential to Editor” section, and submit your "Accept" recommendation.

Reviewer #2: All comments have been addressed

2. Does this manuscript meet PLOS Digital Health’s publication criteria? Is the manuscript technically sound, and do the data support the conclusions? The manuscript must describe methodologically and ethically rigorous research with conclusions that are appropriately drawn based on the data presented.

Reviewer #2: Yes

3. Has the statistical analysis been performed appropriately and rigorously?

Reviewer #2: Yes

4. Have the authors made all data underlying the findings in their manuscript fully available (please refer to the Data Availability Statement at the start of the manuscript PDF file)?

Reviewer #2: Yes

5. Is the manuscript presented in an intelligible fashion and written in standard English?

Reviewer #2: Yes

6. Review Comments to the Author

Reviewer #2: The authors have answered all of my comments in a satisfactory way. Thus, I propose the acceptance of the manuscript.

7. PLOS authors have the option to publish the peer review history of their article (what does this mean?). If published, this will include your full peer review and any attached files.

**Do you want your identity to be public for this peer review?** For information about this choice, including consent withdrawal, please see our Privacy Policy.

Reviewer #2: No
